

# Conversion of Skeletal Society of Radiology annual meeting abstracts to publications in 2010–2015

Daniel Tritz[1], Leomar Bautista[2], Jared Scott[1] and Matt Vassar[3]

[1] Oklahoma State University Center for Health Sciences, Tulsa, United States of America
[2] Department of Radiology, Oklahoma State University Medical Center, Tulsa, United States of America
[3] Department of Psychiatry, Oklahoma State University Center for Health Sciences, Tulsa, United States of America

## ABSTRACT

**Background**. Material presented at conferences is meant to provide exposure to ongoing research that could affect medical decision making based on future outcomes. It is important then to evaluate the rates of publication from conference presentations as a measure of academic quality as such research has undergone peer review and journal acceptance. The purpose of this study is to evaluate the fate of abstracts presented at the Skeletal Society of Radiology Annual Meetings from 2010–2015.

**Materials and Methods**. Conference abstracts were searched using Google, Google Scholar, and PubMed (which includes Medline) to locate the corresponding published reports. The data recorded for published studies included date published online, in print, or both; the journal in which it was published; and the 5-year journal impact factor. When an abstract was not confirmed as published, authors were contacted by email to verify its publication status or, if not published, the reason for nonpublication.

**Results**. A total of 162 abstracts were published out of 320 presented (50.6%) at the SSR conferences from 2010 to 2015 with 59.9% (85/142) of publications occurring within two years of the conference date (not counting abstracts published prior to conference). Mean time to publication was 19 months and is calculated by excluding the 20 (12.3%) abstracts that were published prior to the conference date. The median time to publication is 13 months (25th–75th percentile: 6.25–21.75). The top two journals publishing research studies from this conference were *Skeletal Radiology* and *The American Journal of Roentgenology*. These journals accepted 72 of the 162 (44.4%) studies for publication. Of the 14 authors who responded with 17 reasons for not publishing, the most common reasons were lack of time (7–41.2%), results not important enough (4–23.5%), publication not an aim (3–17.6%), and lack of resources (3–17.6%).

**Discussion**. At least half of the abstracts presented at the annual meeting for the Society of Skeletal Radiology are accepted for publication in a peer-reviewed journal. The majority (59.9%) of these publications were achieved within two years of the conference presentation. The rate at which presentations are published and the journals that accept the abstracts can potentially be used to compare the importance and quality of information at conferences.

Corresponding author
Daniel Tritz, daniel.tritz@okstate.edu

PeerJ ________________________________________________

## INTRODUCTION

A 2010 study on the global burden of disease found that the prevalence of musculoskeletal disorders was 8.4%; however, this rate increased to 20–40% in people over the age of 80 years (*Smith et al., 2014*). To improve outcomes for patients with these disorders, scientific advancements are needed. The dissemination of scientific knowledge occurs through periodic meetings as well as through the publication of research in peer-reviewed journals. The Skeletal Society of Radiology (SSR) works to develop skills and conduct research in musculoskeletal radiology. This society holds an annual meeting to encourage research activity and dissemination, stimulate collaboration among its members, and refine subspecialty expertise through continuing medical education and presentation opportunities. The research abstracts submitted to this meeting undergo formal peer-review and while beneficial, evidence suggests that publication rates for conference abstracts are higher for those with positive findings, randomized designs, and larger sample sizes (*Scherer, Langenberg & Von Elm, 2007*). These studies have also found that oral presentations and those falling into the category of experimental research are also more frequently published (*Easterbrook et al., 1991*; *Dickersin, Min & Meinert, 1992*; *Scherer, Langenberg & Von Elm, 2007*).

In this study, we evaluate the fate of abstracts presented at the SSR (Skeletal Society of Radiology) Annual Meeting from 2010 to 2015. We also analyze the length of time for abstracts to be published from the date of the conference and catalogued the top ten most common journals that publish these studies after presentation. In addition, we explore the reasons for nonpublication to evaluate the most common barriers. To our knowledge, this is the first study examining the publication rates from a radiology or musculoskeletal conference within the United States or for this society's annual meeting.

## METHODS

### Oversight and reporting

This study did not meet the regulatory definition of human subjects research as defined in 45 CFR 46.102(d) and (f) of the Department of Health and Human Services' Code of Federal Regulations and, therefore, was not subject to Institutional Review Board oversight. We applied relevant Statistical Analyses and Methods in the Published Literature (SAMPL) guidelines for reporting descriptive statistics. We developed our methodology by consulting previous studies on the publication rates of conference abstracts (*Hoag, Elterman & Macneily, 2006*).

### Locating conference abstracts

SSR oral and poster presentation abstracts from 2010 to 2015 were located using information from the SSR website about past conference summaries on PubMed. This time period was selected based on The European Society of Skeletal Radiology's decision in 2011 that a duration of two years from research conception to publication is acceptable (*Parkar, Vanhoenacker & Adriaensen van Roij, 2013*). Including abstracts from 2010 to 2015 leaves at least two years for any abstracts to reach publication. The search period for the abstracts

included the period 2009–2017 to allow for any presentations published before the first conference in 2010 and two years to submit projects after the final conference in 2015.

## Search strategy for published manuscripts of conference abstracts

We attempted to locate the published reports of conference abstracts using a predefined search algorithm (Fig. 1). The search algorithm was developed by two investigators and pilot-tested on 25 abstracts. We assessed the optimal order in which to search databases as Google, Google Scholar, and PubMed. We also varied the searches using combinations of keywords and authors' names as well as full title searches to determine which strategy would provide the greatest precision to locate published reports. One investigator first searched the three databases (in the order: Google, Google Scholar, and PubMed) by using the full conference abstract title. If this strategy failed to locate a published report, then the investigator performed the search using the first author's last name and keywords from the abstract. For those abstracts in which no published report could be located, a second investigator repeated the search strategy. These searches were made in June/July of 2017.

## Contacting authors

If no publication had been located after two searches, a standardized email was then sent to the first author of the conference abstract. If contact information for this author could not be located, we then contacted all authors for whom an email address was found (Fig. 1). This email provided authors with the opportunity to comment on whether the study was published and, if so, the reference for the publication. In the event of non-publication, abstract authors were asked to provide a reason for non-publication. The standardized response options for non-publication were based on a systematic review by Song et al. that analyzed 38 survey reports on investigator-reported reasons for non-publication (*Song, Loke & Hooper, 2014*).

Figure 2 provides a step-wise description of the process later discussed in the results. The initial number of papers identified was from the investigators' data collection before contacting the listed authors. Email responses from the authors were either the reasoning why the abstract was not published or a link to the article that was not found previously. The newly located articles are added into the total at the bottom of the figure and the end result.

## Data collection

After locating a published study thought to be a conference abstract, we then compared the author list, methods, and results. If at least two of the following criteria were met, we counted the abstract as published: (1) the results from the published report matched the results in the conference abstract, (2) the methodology from the published report was similar to the methodology described in the conference abstract, or (3) the first author of the conference abstract was included in the author list of the published study.

Data were extracted from the published study using a standardized, pre-specified, and piloted Google Form. We extracted the following information when available: publication title, institution of first author, date submitted to journal, date accepted for publication, date of in-print publication, date of online publication, journal name, and 5-year journal

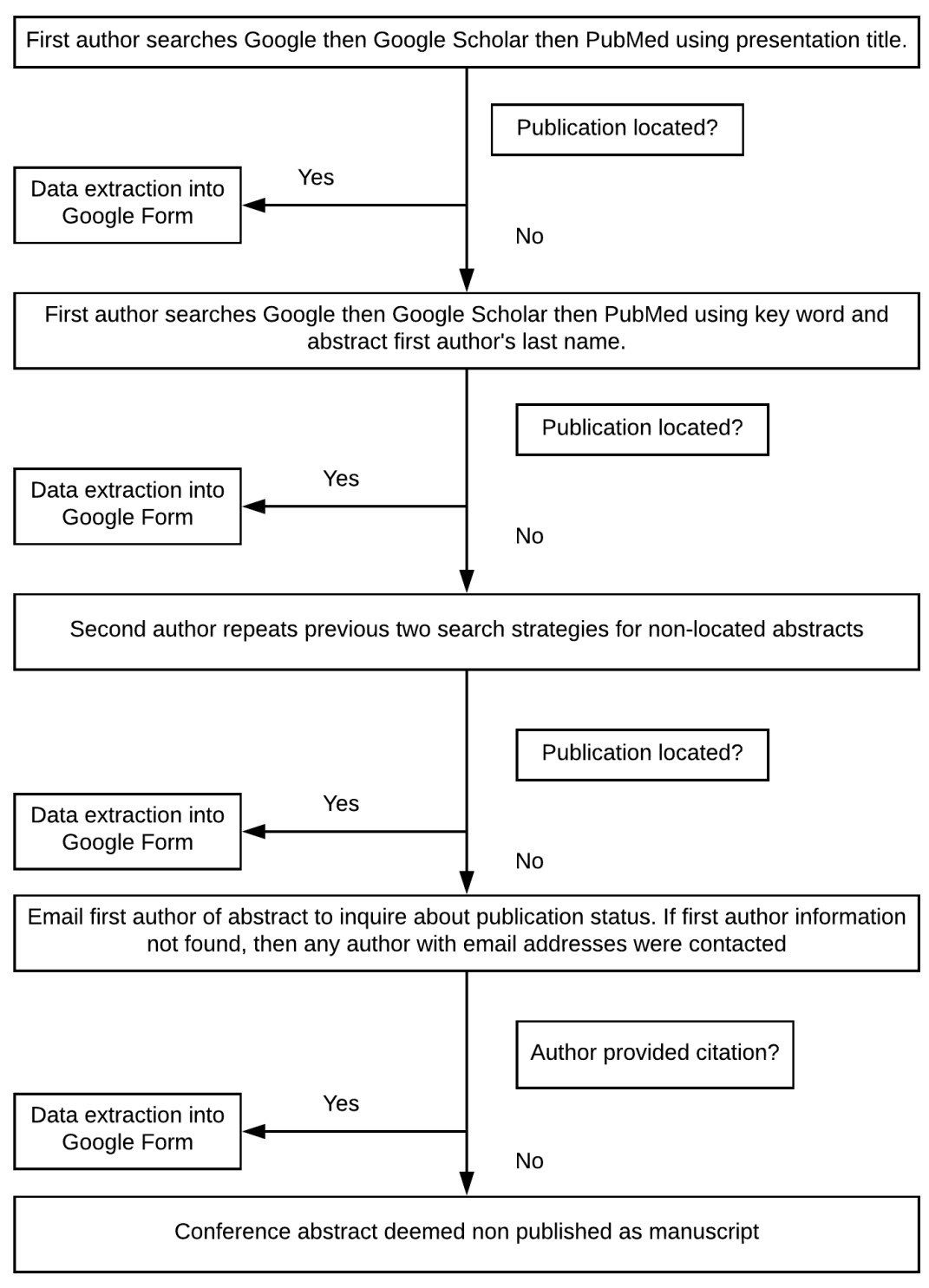

**Figure 1  Search algorithm to find publications.**

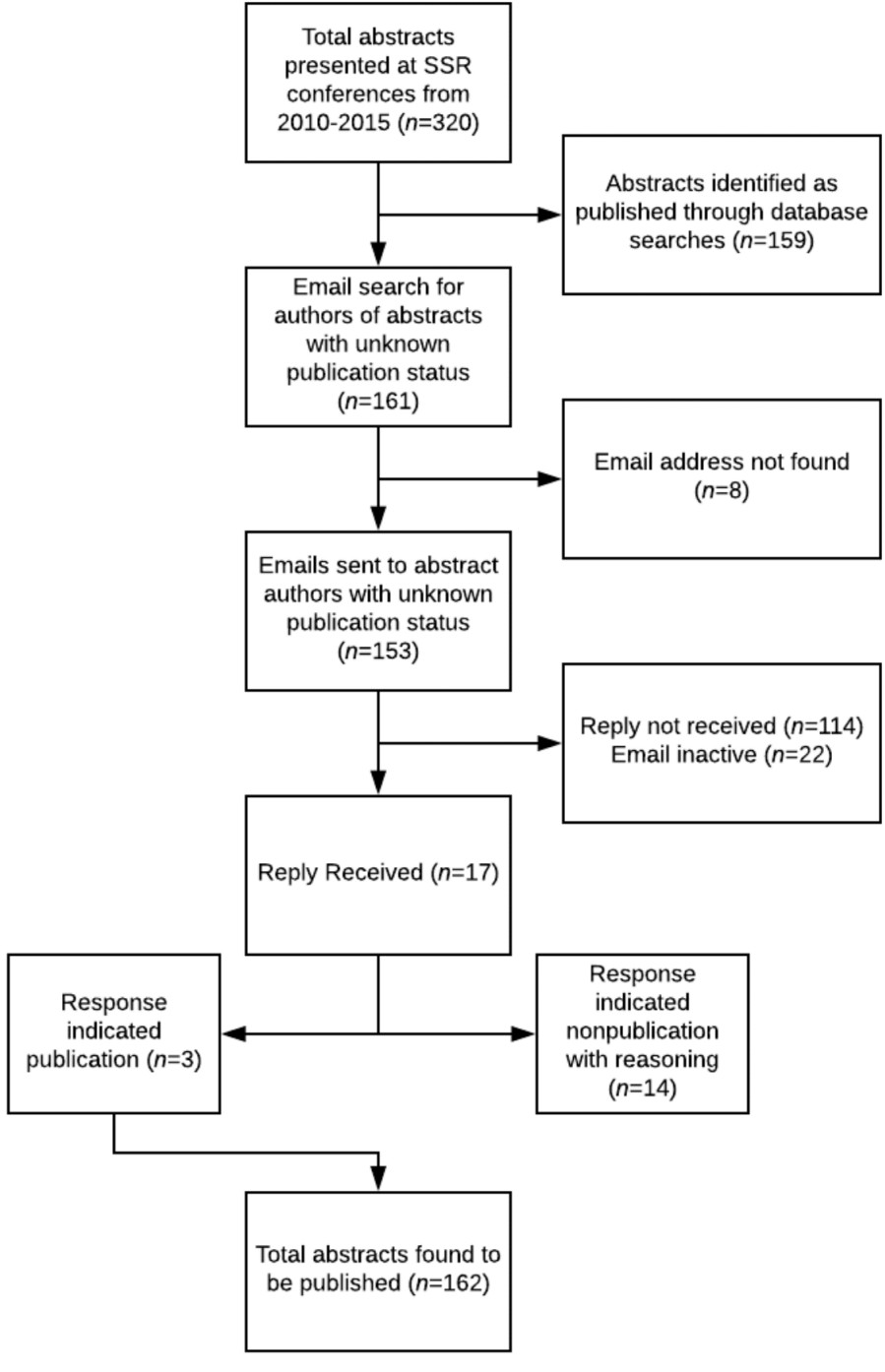

**Figure 2** Flowchart for publication analysis.

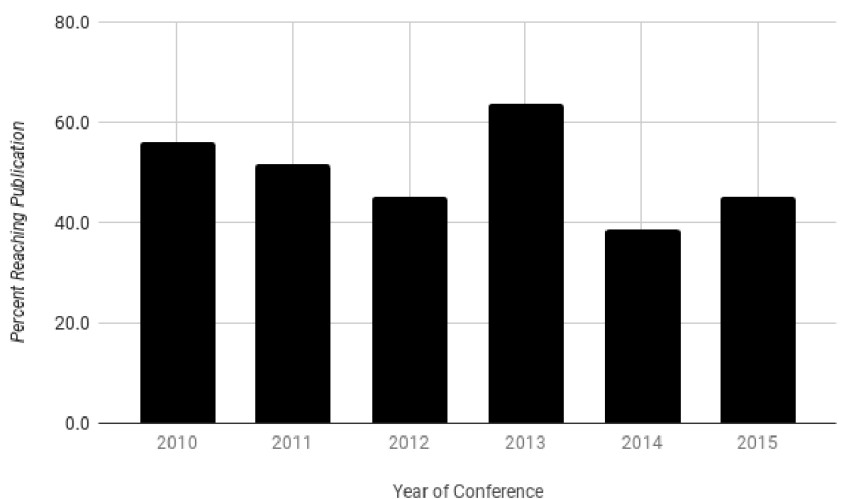

**Figure 3  Percent of published studies by conference year.**

impact factor. We calculated the time to publication based on the number of months between the date of the conference where presented and the publication date online due to the lack of data regarding date accepted for publication. When necessary, we used a zero number of months to indicate studies in which publication occurred prior to conference commencement. The 5-year journal impact factor was located by searching the article name in the "Web of Science" from clarivate analytics.

## RESULTS

From 2010 to 2015, 320 abstracts were presented. Of these, 162 (50.6%) were found to be published in a peer-reviewed journal (Fig. 2). The highest rate of publication by the time of data extraction was in 2013 at 63.6% (28/44) and the lowest was the same in 2012 and 2015 at 45.1% (32/71 and 23/51, respectively) (Fig. 3). Throughout the time period, publication times ranged from 0 (prior to the presentation) to 77 months after the conference date with a median of 13 months (25th–75th percentile: 6.25–21.75). Of the 162 published articles, 20 (12.3%) abstracts were published prior to the conference date when it was presented and are removed from calculations for times to publication. 19 months with 85 (59.9%) of the 142 presentations accepted for publication within two years after presentation. The two years broken down includes 53 (32.7%) within the first year and 52 (32.1%) within the second. A Kaplan–Meier analysis was done to show the proportion of the abstracts which were published measured in months after presentation at the conference (Fig. 4). This analysis includes abstracts that were eventually published and eliminates values of 0 (published prior to conference).

Abstracts from the SSR conference were published in 47 journals. There were 12 journals that accepted three or more abstract presentations for full-text publication (Fig. 5). The top two journals that published abstracts from this conference were *Skeletal Radiology* and

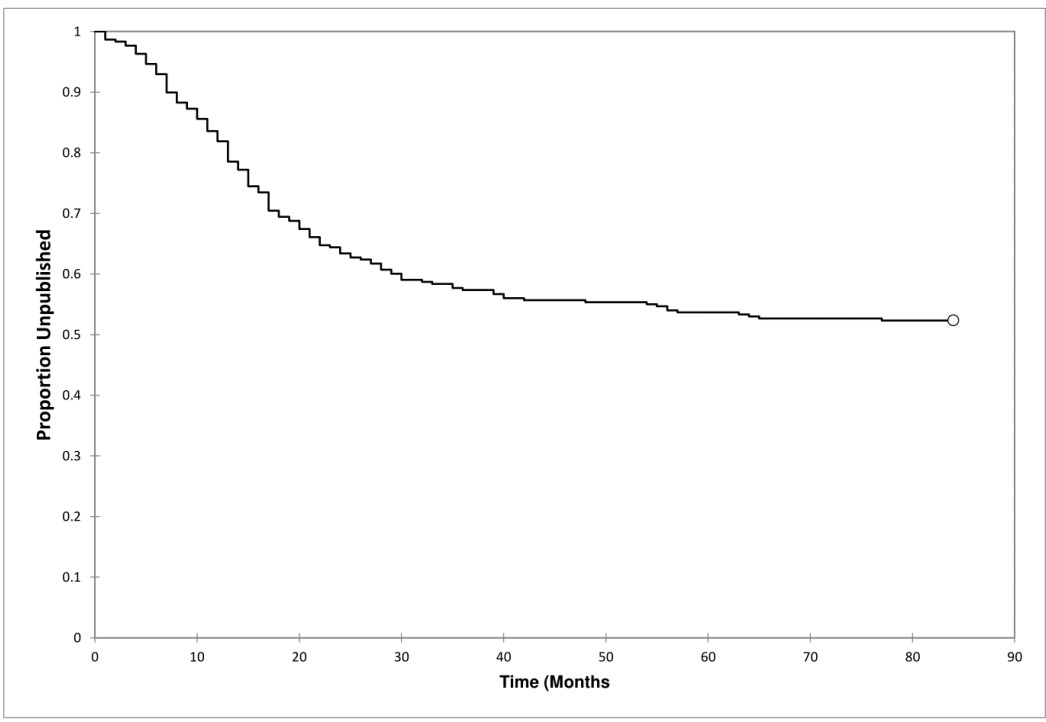

| Time (Months) | 0 | 20 | 40 | 60 | 80 |
|---|---|---|---|---|---|
| Number Unpublished | 300 | 206 | 171 | 162 | 158 |

**Figure 4** **Kaplan Meier analysis excluding abstracts published prior to conference date.** Studies published prior to the conference date were considered 0's and excluded from the analysis as they are not "at risk" for publication.

*The American Journal of Roentgenology. Skeletal Radiology* published 37 of the total 162 (22.8%) and *The American Journal of Roentgenology* accepted 35 of 162 (21.6%).

Of the 320 abstracts, 161 were not found to be published as full-text articles. Email addresses were found for 153 out of 161 presentations. Authors associated with multiple projects were sent one email. A total of 110 unique emails were sent out with 22 being returned as invalid or out-of-date addresses. Seventeen responses were recorded while 114 authors for presentations did not reply. Three responses indicated that the study was published with authorship information and 14 confirmed the paper was not published with 17 reasons why (Fig. 2). The most common barriers to publication were lack of time ($n = 7$–41.2%), results not important enough ($n = 4$–23.5%), publication not an aim ($n = 3$–17.6%), and lack of resources ($n = 3$–17.6%).

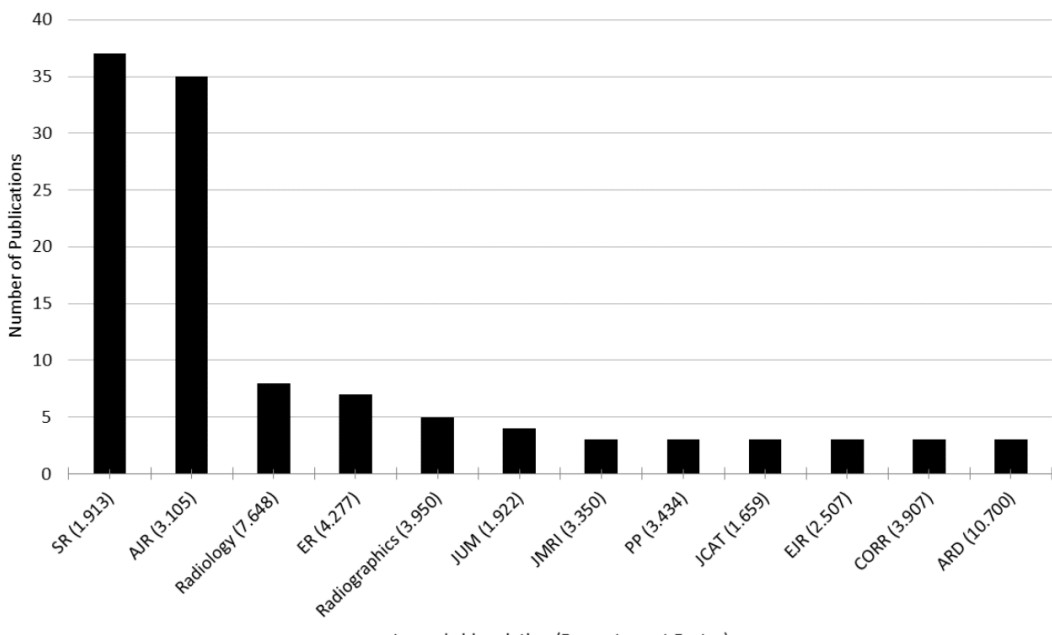

**Figure 5  Journals that published three or more conference abstracts.** SR, Skeletal Radiology, AJR, American Journal of Roentgenology, ER, European Radiology, JUM, Journal of Ultrasound in Medicine, JMRI, Journal of Magnetic Resonance Imaging, PP, Pain Physician, JCAT, Journal of Computer Assisted Tomography, EJR, European Journal of Radiology, CORR, Clinical Orthopaedics and Related Research, ARD, Annals of the Rheumatic Diseases.

## DISCUSSION

The publication rate of SSR conference abstracts from 2010 to 2015 was 50.6%. Scherer et al. evaluated 79 biomedical conferences and reported an overall publication rate of 44.5%, which is lower than the SSR. Because this is the first study of publication rates for a musculoskeletal radiology conference in the United States, comparison can only be made with international meetings. The rate of publication from the SSR conference is favorable compared with that of the Canadian Association of Radiologists for 2005–2011 (28%) (*Dressler & Leswick, 2015*), the Korean Radiological Society for 2001–2002 (23.6%) (*Ha et al., 2008*), and the European Society of Skeletal Radiology for 2008–2009 (45%) (*Parkar, Vanhoenacker & Adriaensen van Roij, 2013*). *Miguel-Dasit et al. (2006a)* and *Miguel-Dasit et al. (2006b)* analyzed abstracts submitted to the European Congress of Radiology (ECR). In 2000, abstracts submitted from United States investigators were more likely to reach publication (62%). This finding was confirmed when extending the conference dates to include 2000–2004 (76%). Loughborough et al. found that publication rates for abstracts in the musculoskeletal category of this Congress were 51%, which is very similar to our findings (*Loughborough et al., 2016*)

Forty-seven journals published 162 abstracts presented at the 2010–2015 SSR conferences. The two most common journals for publication were *The Journal of Skeletal Radiology* and *The American Journal of Roentgenology* with 37 and 35 abstracts accepted,

respectively. *The Journal of Skeletal Radiology* has a 5-year impact factor of 1.913 and is the official journal of the Society of Skeletal Radiology. *The American Journal of Roentgenology* has a higher 5-year impact factor of 3.105 but is not associated with the SSR. The 5-year impact factor of journals that published three or more articles ranges from 0 to 10.700. *Rodrigues et al. (2014)* found that higher impact factors correlate with studies that use the highest level of evidence to support their outcomes. In addition, the impact factor of a journal has been shown to strongly correlate with the newsworthiness and methodological quality of its recent publications (*Lokker et al., 2012*). Although the impact factor of a journal is useful as a binary measurement for comparison in this study, there are many limitations for using it. A journals impact factor can be influenced by a few recent papers being cited many times compared with other publications. This imbalance makes the impact factor a better marker for comparing journals with each other as opposed to individual papers due to varying numbers of citations per paper (*Adler, Ewing & Taylor, 2008*).

The importance of journal impact factors for conferences corresponds to the quality of the presented abstract. Abstracts for the SSR are reviewed and officially accepted by their programming committee, which consists of the President-Elect as Chair, the SSR Secretary, and one Society full member in good standing. There is little information on the criteria used to select presentations other than a focus on correlating abstracts with session topics or program needs and objectives. Blackburn et al. had reviewers use a five-point scale for the overall poster as well as an evaluation of its importance, quality, implications for theory, and implications for practice. This study found that based on the mean scores given for each presentation, 17–20% of subjective decisions for acceptance or rejection would have been reversed (*Blackburn & Hakel, 2006*). Kuczmarski et al. surveyed 27 large scientific conferences to evaluate the screening process for conference abstracts to be presented and found that only two (7%) made the scoring process available to submitters and the public (*Kuczmarski, Raja & Pallin, 2015*). As a follow-up to the survey, they created a comprehensive scoring system that emphasized transparency and objectivity. This scoring system asks reviewers to provide 0 to 2 points for each of the seven categories that are outlined by general criteria for each grade, as well as specific examples. We suggest that the SSR takes steps to ensure transparency in their scoring process that may result in improved abstract quality (*Kuczmarski, Raja & Pallin, 2015*)

The most common reasons that authors cited for nonpublication were lack of time, results not important enough, publication not an aim, and lack of resources. Krzyzanowska et al. found lack of time to be a major barrier to publication in large randomized trials that were presented at the American Society of Clinical Oncology (*Chapman et al., 2012*). The belief that results were not important enough corresponds with the notion that publication is more likely when results are positive or when they correlate with a significant outcome (*Hopewell et al., 2009*). This phenomenon is known as "publication bias" and is a common problem in medical literature. Evidence suggests that editors are less likely to publish negative results, peer-reviewers are less likely to recommend them for publication and researchers are less likely to submit them in the first place (*Dickersin, Min & Meinert, 1992*). This form of bias contributes to overestimated treatment effects because only statistically significant outcomes and large magnitude effect sizes are available to represent

an intervention's effectiveness. The null or negative findings from studies of these same interventions are neither available nor known to the research community as they have not been published. Efforts to minimize publication bias have included alternative peer review models and negative results sections in journals. Such efforts will hopefully contribute to positive changes on this important issue.

This study utilized multiple methods to identify publications from conference abstracts. The searches were made using Google Scholar, PubMed, and Google. These search engines could potentially miss articles published in databases that are not connected. An attempt was made to contact at least one author associated with each presentation through email. Although not all authors replied, the two-researcher search method and attempted contact with authors makes the potential for omissions limited. The conference summaries listed oral presentations for each abstract submitted to the annual conference, but poster presentation information was only listed for two years. During these two years the rate of publication was similar to the overall publication rate; therefore, it was decided to not separate the data just for the two years of posters. Finally, we did not impose time limits in our investigation; abstracts presented in 2010, for example, would have a longer time window to reach publication than abstracts presented at the 2015 conference. Given the Kaplan-Meier analysis (Fig. 4) demonstrating the majority of publications occurring within 2 years, it is likely that a majority of abstracts presented in later years such as 2015 had reached their final fate by our study's end point.

## CONCLUSION

In summary, approximately half of the abstracts presented at SSR Annual Meetings are eventually published. The rate of publication for abstracts presented at the SSR are higher than those analyzed from radiology conferences in other countries like Canada and Korea, and are comparable with the European College of Radiology. The majority of abstracts were accepted for publication by two years after the conference date. Those abstracts that do not reach publication are the result of a lack of time or resources, a perception of unimportant findings, and publication not being an aim of the abstract author. Such reasons are problematic when they contribute to publication bias and efforts should be made to encourage the publication of results, regardless of the strength or direction of study findings. Furthermore, efforts should be made to increase the transparency of the scoring rubrics used by conferences to increase the quality of the submissions they receive.

### Funding
The authors received no funding for this work.

### Competing Interests
The authors declare there are no competing interests.

## Author Contributions

- Daniel Tritz and Leomar Bautista conceived and designed the experiments, performed the experiments, analyzed the data, contributed reagents/materials/analysis tools, prepared figures and/or tables, authored or reviewed drafts of the paper, approved the final draft.
- Jared Scott conceived and designed the experiments, contributed reagents/materials/analysis tools, prepared figures and/or tables, authored or reviewed drafts of the paper, approved the final draft.
- Matt Vassar conceived and designed the experiments, contributed reagents/materials/analysis tools, authored or reviewed drafts of the paper, approved the final draft.

## Data Availability

The raw data are provided in the Supplemental File.

## Supplemental Information

Supplemental information for this article can be found online at http://dx.doi.org/10.7717/peerj.5817#supplemental-information.

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
