# Peer review of "Conversion of Skeletal Society of Radiology annual meeting abstracts to publications in 2010–2015"

_PeerJ, doi:10.7717/peerj.5817_

## Round 0.1 · original submission · Minor Revisions

The reviewers have made a number of, what I regard as, useful and insightful suggestions for your revisions to the manuscript. I invite you to consider each of their suggestions and respond through changes in the manuscript or an explanation of why you do not feel that particular change is required.

Reviewer 1 ·

Basic reporting

This is a well written report. The language is clear.

Experimental design

The methodology for this piece is excellent and superior to the majority of publications on this subject matter in other specailties.

Validity of the findings

The publication rates are higher than observed for other specialties. Could the authors explain why they would think that this might be the case.

Additional comments

is there data on the acceptance rates for abstract for this meeting, This might give context to the difficulty/competitiveness to gain acceptance and therefor may reflect upon the quality of research presented.

For line 177, an alternative term to manpower should be used. In 2018, this term is less acceptable and will likely be less acceptable in years to come. Perhaps say ‘lack of resources’.

For Figure 1, are there numbers for each of these boxes?

Reviewer 2 ·

Basic reporting

Thank you for the opportunity to review this paper. It is well structured, and sufficiently referenced to justify the conclusions.
There are a few areas where I think the language could be improved, for example in the abstract "research that has the potential to be...implanted" could be better stated.
Please use "mean" or "median" rather that "average" throughout the paper, as "average " does not have a precise statistical meaning.
The sentence "The shortest time to publication was in 2015 at 10 months, on average..." is quite confusing. I suggest starting by stating what the overall range of publication times is (over the entire time period), and then comparing years. I would give similar advice for the sentence starting in line 139 in the discussion.

Experimental design

The study used a robust method of gathering data, the research question was well- defined, original and meaningful.

Validity of the findings

The data are generally robust, and the conclusions well- founded.
I would suggest that you use Kaplan- Meier or similar techniques to correct for different and incomplete follow up times between the years- this will allow for more meaningful comparisons between the years that are studied.

·

Basic reporting

1. Consider rewording second sentence of abstract (lines 11-13) as currently too broad a statement. Is publication of studies presented at conferences necessarily a sign of how it impacts upon clinical practice? Publication of conference material suggests it has successfully undergone further (usually more stringent) peer review and potentially of higher academic quality than that which isn’t, but this doesn’t necessarily translate into impact on clinical practice.
2. First sentence in abstract results (line 21) – either use prose or numerical digits but don’t use both, particularly in the same sentence. This applies throughout the paper.
3. Abstract discussion does not reflect abstract results (no mention in results of publication rates from other conferences for objective comparison).
4. Sentence 41-44 needs a reference as it is describing (presumably published) evidence.

Experimental design

1. The authors should be commended for their thorough methodological design in determining if abstracts have been published, including contacting the presenting authors. However although Firgure.2 explains the method for determining publication, it is unclear from the text that if the situation occurred whereby no abstract was found with traditional search engines and the author failed to contact the study group (‘reply not received’ in the flowchart), whether the abstract was deemed non published (presumably so). Additionally in this flow chart, the initial figure after search engines is n=161, but the final ‘total publications’ is n=162. This number in the final box is inappropriate as various papers have been removed prior to this point.
2. Lines 61-69 – the dates of the search period for published abstracts should be clearly described. Presumably it is all abstracts published between 2010-2017?
3. Line 100 – how did the authors determine impact factor of journals?
4. Lines 100-102 – it is unclear if the time to publication is from the original conference to the publication or from 2010 to the publication (eg if a conference abstract presented in 2013 was published in 2015, would the time to publication be 2 or 5 years?). However, see comment 3.2.
5. On a similar note, it is unclear if each year of abstracts had a 2 year limit to publication, or were abstracts presented in 2010 given the same time window as abstracts from 2015 (i.e. an extra 5 years?). If the latter, this will bias results in favor of the earlier conferences in terms of publication percentage and should be described.

Validity of the findings

1. Consider using the following three studies when comparing to publication rates from ECR;

• Miguel-Dasit A, Martí-Bonmatí L, Aleixandre R, Sanfeliu P, Bautista D. Publication of material presented at radiologic meetings: authors’ country and international collaboration. Radiology 2006;239(2):521–528.
• Miguel-Dasit A, Martí-Bonmatí L, Sanfeliu P, Aleixandre R. Scientific papers presented at the European Congress of Radiology 2000: Publication rates and characteristics during the period 2000-2004. Eur Radiol 2006;16(2):445–450.
• Loughborough W, Dale H, Wareham JH, Youssef AH, Rodrigues MA, Rodrigues JCL. Characteristics and trends in publication of scientific papers presented at the European Congress of Radiology: a comparison between 2000 and 2010. Insights Imaging 2016;7(5):755–762.

2. Some of the discussion should be within methodology or results. For example, line 140-143 should be within results and effectively answers reviewers comments 2.4 and 2.5. Equally, 144-146 should be within the results section but comparison to the Canadian data is appropriately within discussion.

3. It is unclear what the numbers in brackets refer to in line 149 (37) and line 150 (35).

5. Lines 154-158 – consider discussing the limitations of purely using impact factor of a publishing journal to determine the credibility of a study. Whilst in a study such as this, impact factor is a useful (arguably the best) binary, objective measure for assessing scientific quality of studies, there are limitations. For example the fact that it is estimated that the top 20% papers in a journal receive 80% citations (See - http://www.editage.com/insights/why-you-should-not-use-the-journal-impact-factor-to-evaluate-research), therefore publication in a high impact factor journal does not mean the paper will receive a high number of citations.
6. Lines 162-175 – the discussion on limitations and lack of transparency in selecting abstracts for conferences is commendable.
7. Lines 179-191 – the authors should be commended for the discussion around publication bias and their efforts in suggesting ways to overcome this.
8. Lines 197-198 – it is unclear what ‘presented orally for all six years’ means? Was the same abstract presented orally for six years running? Or were oral abstracts presented online for six years following the conference? It currently reads like the former.
9. Conclusion is overall well written but little is made of the binary results of publication rates. Even if not directly discussed, authors should consider alluding to publication rates vs other conferences, length of time to publication and publishing journals as this is a central part of results.

---

## Round 0.2 · Minor Revisions

Thank you for your revisions. All reviewers who have had a chance to respond are happy with the revised version of your manuscript. I have carefully read over the manuscript and made a number of suggestions for you to consider. Many of these are copyediting and most of the rest are simply proofreading. I don’t imagine that these will present you with any difficulty in addressing or explaining why you do not wish to make the suggested change(s). I look forward to your responses to each point and I hope to be able to accept your revised version of the manuscript then.

Line 26: Looking at Figure 6, I think these are 5-year impact factors and I’d make that clear from the outset if this is correct.

Line 31: The mean here, I think, includes the zeros and, if so, I would suggest either using a median to avoid the influence of using zeros versus negative values or, and this would be my strong recommendation, breaking this down into two steps: n(%) already published at the time of the meeting and median or mean only for the remaining abstracts that were published. If the mean already reflects only the post-meeting publications, this could be made clear here. Note that a mean can be useful in such cases but no longer preserves the interpretation of being the typical value, which a median more naturally reflects. It might be useful for you to calculate both the mean and median times and consider whether one or both should be shown here and elsewhere.

Lines 31–32: Similarly, it’s not completely clear at this point whether those abstracts published before the meeting are treated as “within 2 years” (or how many of these there might be). I’d assume so given the information already provided before this point, but have this disconfirmed later on Line 143. Note that including the already published abstracts in the denominator is unfair. Even if all unpublished abstracts were published within 2 years, as it appears to be calculated, the result would still not be 100% (due to the 20 not eligible to be included in the numerator being included in the denominator, giving an upper limit of 87.7%).

Line 35: Could you specify the number of authors who responded here (14 from Line 160)? Otherwise, it’s difficult to interpret the following reasons beyond relative magnitudes.

Lines 36–37: I appreciate that it’s not difficult to calculate if the number of responding authors is provided, but also providing percentages of those responding here would be a small kindness to some readers.

Line 39: What you know at this stage is that 50.6% of the abstracts from 2010–2015 have appeared in your searches of journal publications. While I wouldn’t expect a substantial increase, some of those abstracts have had less time to be published (and some quite a bit more). I wonder if “At least half…” would therefore be a safer statement and one that is guaranteed to remain true in the future.

Line 41: The wording “of the conference being presented at” seems awkward to me and perhaps “of the conference” would suffice?

Line 101: The final date for checking the publication of the abstracts should be included around here. I appreciate that publications from 2009–2017 (Line 88) were included, but databases are sometimes slow to add articles and so this date is important for theoretical replication of the results.

Line 105: I wasn’t sure if “any author” here meant that “all authors” with an email address were contacted in this case, or if “a randomly chosen author” with an email address, or the “next listed author” after the first author with an email address. Sorry for the pedantry!

Lines 126 and 130: As with Line 26, looking at Figure 6, I think these are 5-year impact factors and I’d make that clear here too if this is indeed the case.

Line 137: The available time varies by meeting year and I wonder if it would be useful to remind the reader of this here by adding “by the time of data extraction” or “by the end of 2017” between “publication” and “was” here, or something similar.

Lines 139 onwards: I think the order of these results needs rethinking as we get the lower limit of 0 (pre meeting) on Line 139, then the mean on Line 140 (which may or may not include the zeros, I’m still note confident on this point), and then the number of zeros on Lines 142–143, and finally those achieving the two-year guideline on Line 143. I appreciate that the section is short and not terribly confusing, but perhaps breaking out the zeros first and then giving the range for the remaining, the mean/median of the rest overall, and then the 2-year (explicitly excluding the “zeros” with this structure so that 100% is theoretically possible even with pre-meeting publications) might flow better. This is just a general suggestion and I’ll make more specific comments below.

Lines 140–141: Including zeros in this calculation would certainly be open to question (see also the comment for Line 31). If the mean is only for those with non-zero values, all that would need to be done here, assuming you don’t wish to switch to medians, would be to make this clear.

Lines 141–142: This result follows from the nature of the data (only earlier meetings could generate long publication times and for more recent meetings, only publications with shorter times could be included) and so is a naturally biased comparison. I’d suggest deleting this sentence unless there is something here I’m missing.

Lines 161–162: Again, as a kindness to your reader, you could add percentages here. Did any authors indicate multiple reasons; if so or if not, this should be made clear here.

Lines 165–167: While I appreciate that most abstracts appear to be published within that 2 year window, over one-fifth (22.8%) of abstracts that were published had this event occur more than 2 years after the meeting. I think interpreting this is difficult without knowing the median years of follow-up as the publication rate can only remain the same or, more likely over periods up to at least 5 years and possibly longer given your 77 month publication, increase with the passage of time (the median follow-up would be roughly 5 years I think for you assuming midyear for meeting and extraction at the start of this year). The same addition of follow-up times would be useful for the results on Lines 169–178. At least, you need to give your median follow-up time and note whether the other studies had similar, shorter, or longer periods.

Line 185 onwards: Note that these are 5 year impact factors if appropriate.

Line 187: Apologies for the ignorance, but what does “n.d.” mean here? I’m aware of it being used in references for “no date” but that doesn’t seem to be the case here.

Line 189: I’d give impact factors to 3 decimal places in all cases.

Line 195: You could add “recent” between “few” and “papers” here.

Lines 197–198: Citations is a count noun so use “varying numbers of citations”.

Line 208: Missing space after “reversed”.

Figure 5: The y-axis should read “Proportion unpublished” not “Proportion published” (or you could instead use a K-M failure graph so that the curve moves upwards indicating publication) and should omit the “(%)” as these are proportions. This graph suggests to me that the “zero”s were excluded (all statistical software that I’m aware of will do this, with or without a warning, as survival analysis only applies to those “alive” at baseline and so having the event occurring at a time greater than 0) and again, this point should be made clear (e.g. “excluding abstracts already published before the meeting” or similar). The usual use of a K-M curve would be to show cases without an event at follow-up (so the curve does not reach zero unless the event has occurred to all) and if unpublished abstracts have been excluded, this needs to be clear in the figure title. Otherwise, you could include all abstracts here in the graph except for those published before the meeting. This looks like a Stata graph, and adding a risk table (“,risktable” in Stata) might help the reader to understand the number of abstracts still “at risk” (survival analysis terminology isn’t terribly appropriate here) of publication. It could be worth looking at in any case.

Figure 6: The journal names are a mixture of upper case (e.g. Skeletal Radiology) and title case (e.g. Radiology). I would suggest using the latter to slightly reduce the space taken up by the journal names, or using abbreviations on the x-axis and provide a key to those abbreviations in the caption. Note also, that all of the five-year impact factors will be to three decimal places and sometimes a trailing zero is missing (e.g. Radiographics). The fact that the values shown are “5-year impact factor”s should be made clear rather than just “Impact Factor”. You could also simplify/shorten the title if you wanted by replacing “with greater than or equal to 3” with “with at least 3” or “with 3 or more”.

Reviewer 1 ·

Basic reporting

The authors have adequately addressed all reviewer comments

Experimental design

The authors have adequately addressed all reviewer comments

Validity of the findings

The authors have adequately addressed all reviewer comments

Additional comments

The authors have adequately addressed all reviewer comments

Reviewer 2 ·

Basic reporting

Thank you,
All the concerns/ recommendations made on the previous review have been adequately addressed in the revised article.

Experimental design

Thank you,
All the concerns/ recommendations made on the previous review have been adequately addressed in the revised article.

Validity of the findings

Thank you,
All the concerns/ recommendations made on the previous review have been adequately addressed in the revised article.

Additional comments

Thank you,
All the concerns/ recommendations made on the previous review have been adequately addressed in the revised article.

---

## Round 0.3 · Minor Revisions

Thank you for your revisions. At the risk of being painful, there are a very small number of copyediting comments regarding the new text you’ve added that I’ll make here along with some clarifications about my suggestions for the Kaplan-Meier curve and then I’ll be very happy to recommend accepting your manuscript.

Abstract, results: “(Interquartile range: 6.25 - 21.75).” Technically, the IQR is the difference between the 75th and 25th percentiles (see https://en.wikipedia.org/wiki/Interquartile_range). You could change this to either “(Interquartile range: 15.5).” or “(25th – 75th percentile: 6.25 - 21.75).” The same applies to the 5th line of the 1st paragraph in the results section of the manuscript itself

Abstract, materials and methods. The text here is not quite the same in interpretation as the newly added text about contacting authors (the new text reads: “If contact information for this author could not be located, we then contacted all authors for whom an email address was found (Figure 1).”). Should “When an abstract was not confirmed as published, an author was contacted by email to verify its publication status…” read instead “When an abstract was not confirmed as published, authors were contacted by email to verify its publication status…”? Related to this, the fourth box in Figure 1 seems inconsistent with the new text around emailing authors (this is specifically about the first author only).

The following two are very minor typographical issues that we might as well fix now:

On the 8th line of the 1st paragraph of the results section, there is a missing space in “19 months with85 (59.9%) of the”

In the last line of the same 1st paragraph, in “eliminates values of o”, I’m assuming this should be “0” (zero) rather than a lower-case letter “o”.

Sorry for not noticing this before, but Figure 1’s top box reads “First author searches Google then Google Scholar ten PubMed using presetation title” (the “ten” and “presetation” are typos here).

I see you’ve also added the number “at risk” to the Kaplan-Meier curve (Figure 4) as I suggested, which I think is useful, but I may not have been entirely clear in my other comments about this figure last time, including: “The usual use of a K-M curve would be to show cases without an event at follow-up (so the curve does not reach zero unless the event has occurred to all)”. The curve should include as censored those presentations that did not achieve publication during follow-up, so the survival curve would not go all the way to the x-axis (i.e. zero surviving) so as to reflect the 158 (~50%) not published at their final follow-up time including the 45% or so not published with seven years of follow-up. There would, thus, still be presentations “at risk” of being published after seven years. Note also my comment from last time that you “should omit the ‘(%)’ as these are proportions”; here you could either remove the reference to percentages on the y-axis title or change the y-axis to range from 0 to 100 and refer to this as “Percentage” rather than “Proportion”. Finally, the previous x-axis title (“Time (Months)”) was more informative in that it gave the units on this axis (compared to “analysis time” now) and “(months)” or similar should be included in whatever title you choose to use for this axis.

There's not much to do in the above and I look forward to being able to accept your revised version soon.

---

## Round 0.4 · accepted · Accept

Thank you for these final revisions. I am now happy to recommend acceptance of your manuscript. Well done on producing what I think is a nice addition to the ever-growing body of work on the translation of conference presentations to journal articles and a very valuable discussion of how this applies in radiology.